# Enhancing Herpes Zoster Vaccination Rates Among Oncology Outpatients: Impact of an In-House Vaccination Initiative [note 1]

**DOI:** 10.3390/cancers17213502

**Published:** 2025-10-30

**Authors:** Alberto Giuseppe Agostara, Silvia Della Torre, Sara Di Bella, Michela Pelliccione, Paola Candido, Valeria Smiroldo, Davide Toniolo, Francesca Zannier, Roberto Bollina

**Affiliations:** 1Department of Oncology, Ospedale G. Salvini—Rho—ASST Rhodense, Rho, 20017 Milan, Italysara.dibella@asst-ovestmi.it (S.D.B.); rbollina@asst-rhodense.it (R.B.); 2Department of Oncology and Hemato-Oncology, Università degli Studi di Milano, 20122 Milan, Italy

**Keywords:** herpes zoster, recombinant zoster vaccine, Shingrix^®^ oncology patients, vaccination rates, quality improvement, immunocompromised patients

## Abstract

**Simple Summary:**

Cancer patients are more likely to develop shingles, a painful skin rash caused by the reactivation of the chickenpox virus. This risk increases because cancer and its treatments can weaken the immune system. A new vaccine, called the recombinant zoster vaccine, can effectively prevent shingles and its complications, but vaccination rates among cancer patients remain low. To address this problem, our oncology center introduced an in-house vaccination program that made the vaccine easily available to patients during their routine cancer care visits. This initiative greatly increased vaccination rates and was well accepted by patients. The results show that offering the vaccine directly within oncology services can remove practical barriers, improve patient protection, and serve as a model for other cancer centers aiming to enhance preventive care for their patients.

**Abstract:**

**Background:** Herpes zoster (HZ) poses significant risks to immunocompromised individuals, particularly cancer patients receiving systemic therapies. The recombinant zoster vaccine (RZV, Shingrix^®^) provides strong and durable protection against HZ and its complications. Nevertheless, vaccination coverage remains low, mainly due to limited awareness among patients and healthcare providers and logistical barriers to vaccine access and delivery. **Materials and Methods:** We conducted a single-center quality improvement (QI) project to enhance RZV uptake among oncology outpatients receiving systemic therapy. Following the Plan–Do–Study–Act (PDSA) model, baseline HZ vaccination coverage was assessed, and an in-house vaccination campaign was implemented. Vaccination rates were monitored every two months over a 14-month period. **Results:** At baseline, only 5.4% (24/446) of patients had received RZV. After 14 months, 365 patients were evaluated for vaccination: 200 (55%) were vaccinated, 134 (37%) were ineligible, and 31 (8%) refused RZV. The overall vaccination rate increased from 5.4% to 44%. Reported adverse events were mild and primarily local reactions, confirming the vaccine’s favorable safety profile in this population. **Conclusions:** This real-world QI initiative demonstrates that an in-house vaccination strategy embedded within oncology services can substantially improve RZV coverage and patient engagement. The approach highlights the key role of oncology teams in leading preventive interventions for immunocompromised patients. By integrating vaccination into routine cancer care, institutions can overcome traditional organizational barriers and align with current ASCO and ESMO recommendations for comprehensive patient protection.

## 1. Introduction

Herpes zoster (HZ), commonly known as Shingles, is a reactivation of the varicella-zoster virus (VZV), which remains dormant in the dorsal root ganglia after an initial varicella (chickenpox) infection [1]. When reactivated, VZV causes a painful, blistering rash that typically follows a dermatomal distribution [2]. Cancer patients are particularly susceptible to HZ reactivation due to their compromised immune systems, a result of both their underlying disease and its treatments, such as chemotherapy and radiotherapy [3,4]. Complications of HZ, such as postherpetic neuralgia, viral dissemination, and secondary bacterial infections can significantly diminish quality of life. While in the general population, an incidence rate of HZ ranging from 5.23 to 10.9 cases per 1000 person-years is reported, among patients with solid tumors, the incidence of HZ is approximately 15 cases per 1000 person-years, with a significant relative risk of infection (relative risk 2.17; 95% confidence interval [CI] 1.86–2.53) [5,6]. The risk is notably higher among those receiving chemotherapy [3,7] and radiotherapy [8,9], while data on immunotherapy remain ambiguous [10,11,12].

Preventive strategies have advanced in recent years with the approval of a non-live recombinant zoster vaccine (RZV, Shingrix^®^, GlaxoSmithKline, Wavre, Belgium) by the FDA in 2017 and the EMA in 2018 [13]. On 23 July 2021, the Food and Drug Administration approved RZV for the prevention of HZ in adults who have or might have an increased risk of HZ due to immunosuppression [14].

Recent evidence showed RZV efficacy remains high at 82.0% eleven years after initial vaccination in individuals ≥50 years [15]. Additionally, this vaccine has significantly altered the preventive landscape for immunocompromised individuals, including cancer patients [16,17]. Italy’s National Vaccine Prevention Plan (PNPV) for 2023–2025 includes Shingrix^®^, offering it actively to individuals aged 65 and older, as well as at-risk individuals aged 50 and older with frailty and chronic conditions. RZV use is recommended in onco-hematologic patients undergoing active treatment or in remission by the main scientific institutions: Associazione Italiana di Oncologia Medica (AIOM), European Society for Medical Oncology (ESMO) [18] and American Society of Clinical Oncology (ASCO) [19]. However, vaccination should be evaluated on a case-by-case basis, considering the patient’s risk profile [18].

In the Italian National Vaccine Prevention Plan (PNPV), a target coverage of 50% was set for specific cohorts, including those aged 65 and older, and individuals over 50 with comorbidities. However, several analyses have shown that the actual vaccination coverage falls significantly short of this goal, with rates ranging from 1.7% to 33.8% [20,21,22].

Although the COVID-19 pandemic has heightened social awareness about vaccinations, several barriers continue to impede proper vaccination among oncology patients. These barriers include a lack of provider and patient awareness, insufficient healthcare provider recommendations, poor patient education and misconceptions about vaccine safety and efficacy, limited vaccine accessibility, and inadequate clinic organization [23]. Additionally, there is a lack of a universal and reliable immunization record.

Recent quality intervention (QI) studies have aimed to increase vaccination rates among immunocompromised and oncology patients. Various strategies have been investigated, including educational programs for both patients and healthcare providers, implementation of electronic reminder systems, and enhanced access to vaccination services. These tailored approaches highlight the ongoing need to improve vaccination coverage in these vulnerable populations [24].

The aim of our QI initiative is to increase the HZ vaccination rate among oncology outpatients. By analyzing patients’ awareness and factors influencing vaccine hesitancy regarding HZ and RZV, we developed an effective in-house vaccination protocol integrated into the clinic’s workflow processes. This is an ongoing project, and we have already presented preliminary data demonstrating its initial impact [25].

## 2. Materials and Methods

### 2.1. Context

This quality improvement (QI) project was a single-center study conducted at the Medical Oncology Department of ASST Rhodense in Milan, Italy, serving a referral base of approximately 500,000 people. During the COVID-19 pandemic, our oncology center served as a vaccination hub for cancer patients, playing an integral role in the regional vaccination strategy. Vaccinations for COVID-19, influenza, and HZ are managed at the local health authority level with guidance from general practitioners.

### 2.2. Setting and Study Population

A multidisciplinary QI team was assembled, consisting of 11 oncologists, 14 oncology nurses, two pharmacists, and a QI specialist. The team collaborated to identify the root causes of the problem, develop and implement interventions, and track project progress. The study population consisted of patients with solid tumors receiving active systemic treatment at the outpatient medical oncology center. This study followed the four-step Plan-Do-Study-Act (PDSA) iterative process to address the low vaccination coverage for HZ among oncology patients (Figure 1). The PDSA cycle aimed to continuously improve the process of increasing HZ vaccination rates among oncology patients, ensuring that interventions were data-driven and responsive to patient needs and preferences.

The Plan-Do-Study-Act (PDSA) cycle is a four-step iterative process used for continuous improvement in various fields, including healthcare, manufacturing, and business. It is designed to test changes in real-world settings and implement improvements based on observed results.

#### 2.2.1. Plan Phase

In the planning phase, the multidisciplinary QI team conducted a comprehensive analysis of vaccination strategies for oncology patients, leveraging insights from the COVID-19 vaccination efforts during the pandemic. The team reviewed the current literature on the importance of vaccinations for viral diseases such as COVID-19, influenza, and HZ, emphasizing the significant risks these diseases pose to the vulnerable oncology population. To facilitate the implementation of the QI program, several staff education meetings were organized. These targeted meetings aimed to identify the main barriers to vaccination among oncology patients, including inadequate healthcare infrastructure and low awareness. The sessions focused on understanding how these limitations affect vaccination uptake and developing strategies to overcome them.

#### 2.2.2. Do Phase

During the “Do” phase, the first QI step involved administering a rapid survey from March 2023 to May 2023 to the entire oncological cohort of 446 patients. This survey was completed anonymously on paper and included six items focusing on the vaccination rates of HZ, COVID-19, and influenza, as well as previous HZ and COVID-19 infections among oncology patients. The surveys were distributed to patients by their doctors during medical visits and collected in designated boxes located throughout the oncology day hospital. The survey aimed to provide a snapshot of vaccination adherence within a real-world cohort of oncology patients.

#### 2.2.3. Study Phase

In the “Study” phase, the questionnaire results were analyzed to assess the baseline vaccination adherence of the oncology cohort. A subsequent meeting of the multidisciplinary QI team was organized to discuss the results and evaluate any discrepancies between the actual vaccination rates of the patients and the target coverage rates established by the PNPV.

#### 2.2.4. Act Phase

In May 2023, an in-house vaccination program was launched at the Oncology Day Hospital. The vaccination protocol was aligned with the recent position paper by AIOM. During scheduled oncology visits, the recombinant zoster vaccine (RZV) was offered to eligible patients undergoing active medical oncology treatments (chemotherapy, immunotherapy, targeted therapy, endocrine therapy, or combined treatments). Patients received comprehensive information about the benefits and potential adverse reactions of RZV.

Patients with a poor clinical condition (ECOG Performance Status > 2), a life expectancy of less than three months, or active acute HZ were excluded. For patients undergoing intravenous therapy, the vaccine was not administered on the same day as their infusion treatment. However, for patients receiving oral therapy, the vaccine was administered without pausing their medication. For patients with a history of allergic reactions to vaccines, an allergist evaluation was requested. No other vaccinations were administered simultaneously with RZV.

The vaccine was given in two doses of 0.5 mL each, injected into the deltoid muscle at month 0 and month 1 or 2, according to the summary of product characteristics (SmPC) and each patient’s medical condition. In Italy, the recombinant zoster vaccine (RZV) is provided free of charge to individuals aged ≥65 years and to immunocompromised adults aged ≥ 50 years, including oncology patients, according to the National Vaccine Prevention Plan (PNPV 2023–2025). An informational brochure detailing RZV indications, efficacy, and safety was available in the oncology day hospital. After RZV administration, patients were provided with a paper-based diary to record any adverse events following immunization (AEFIs) for up to seven days after vaccination. They were instructed to complete this form. During the second visit, patients were asked to return the diary and report any occurrences of HZ during the interval.

### 2.3. Outcomes

The primary endpoint of this study was to evaluate the increase in the vaccination rate for RZV among oncology patients before and after the quality improvement (QI) interventions. Post-intervention vaccination rates were reassessed every two months. Secondary outcomes included the incidence of adverse events following immunization (AEFI) and the overall feasibility of integrating the vaccination program into routine oncology care. This comprehensive assessment aimed to provide valuable insights into the impact of a structured vaccination campaign on improving RZV uptake among oncology outpatients.

### 2.4. Statistical Analysis

Continuous quantitative variables were expressed as mean ± standard deviation, discrete quantitative variables as median, and qualitative variables as percentage (proportion). Vaccination rates were calculated as the ratio between the number of vaccinations and the total number of patients at each specific time point. The vaccination rate was assessed at baseline, every two months, and at the end of the observation period (May 2023–June 2024). To obtain a more accurate estimate, the total number of oncology patients on active treatments was recalculated every two months. This total included vaccinated patients, patients who refused the vaccine, patients deemed clinically ineligible for vaccination by the oncologist, and patients not yet assessed.

### 2.5. Ethical Considerations

This quality improvement initiative did not undergo ethical review by an institutional ethics committee because the administration of the Shingrix vaccine was carried out in accordance with established clinical guidelines for patients with medical indications for the vaccine. The hospital-based vaccination center was established to provide structured and accessible vaccination services for oncology patients, aligning with standard medical practices and prioritizing patient safety and well-being. All participating patients provided informed consent, and the initiative was implemented with full transparency regarding the benefits and potential risks associated with the vaccination.

## 3. Results

### 3.1. Survey (“Do” Phase)

Between March 2023 and May 2023, 446 oncology outpatients from our department completed the baseline questionnaire. At the time of survey completion, all patients were receiving systemic oncological treatment. Of these, 24 patients had already been vaccinated with RZV (5.4%), while 6.5% had experienced at least one episode of HZ in the past. Vaccination rates for COVID-19 and influenza were 97.5% and 75.6%, respectively (Table 1).

### 3.2. In-House Vaccination (“Act” Phase)

#### 3.2.1. Sample Description

Over a 14-month period from May 2023 to June 2024, 200 patients received the RZV at our clinic through the in-house vaccination campaign. The main demographic characteristics of the patients are summarized in Table 2. The patients had a median age of 71 years (range 42–93; SD = 10). Women accounted for 58.5% of the sample and men for 41.5%. The median age was 70.5 years (range 42–93; SD = 10.34) for women and 71 years (range 47–93; SD = 9) for men. Sixteen patients had experienced an HZ infection. The most common cancers were breast (27.5%), lung (22%), colorectal (13.5%), and prostate (9%). The majority of patients had metastatic disease (64%), with 110 patients receiving first-line treatment and 26 patients on subsequent lines of therapy. All patients were undergoing systemic treatments. Additionally, 39 patients had received radiotherapy within the two years prior to the first dose. Fifty-nine patients (29.5%) were receiving chemotherapy, with 15% and 45% on high- and intermediate-risk regimens for febrile neutropenia, respectively. Seventy-nine patients (39.5%) were on oral targeted therapy during vaccination, while 62 patients (31%) were receiving immunotherapy, with 9% of these in combination with chemotherapy. Regarding non-oncological comorbidities, 79 patients (39.5%) had at least three comorbidities, 25 patients (12.5%) were on chronic steroid therapy, and 70 patients (35%) were on polypharmacy regimens involving at least five medications. RZV was administered on average seven days before (range 1–51) and eight days after (range 1–70) intravenous therapy, respectively.

#### 3.2.2. Outcomes

At the start of the in-house vaccination program, 446 patients constituted the cohort of oncology patients undergoing active treatment at our department. The initial vaccination rate was 5.4% (*n* = 24).

The total number of patients on active treatment, vaccinated patients, vaccine refusers, ineligible patients, and those not yet assessed were calculated every two months for limiting the impact of new and lost patients. (Figure 2). Over the study period, 200 patients received at least one dose of RZV, resulting in a vaccination rate of 44% at the last evaluation (June 2024). The average bi-monthly increase in the number of newly vaccinated patients was 28.6 (range 12–44).

In total, 365 patients were evaluated for RZV over the 14 months. Of these, 200 patients were vaccinated, 134 were deemed ineligible for vaccination, and 31 refused RZV. The proportion of patients not yet assessed for vaccination decreased from 79.5% to 35.7% by the final evaluation. In contrast, 15.1% of patients were deemed ineligible for vaccination, and 5.2% refused the vaccine. Among the 31 patients who refused vaccination, the main reasons were fear of adverse effects (*n* = 14, 45%), belief that vaccination was unnecessary or potentially interfering with cancer therapy (*n* = 10, 32%), and general vaccine hesitancy or misinformation (*n* = 7, 23%).

From the total vaccinated cohort, 175 patients received two doses of RZV, while 25 patients received only one dose. Of the latter, 18 patients are scheduled for their second dose, seven did not receive the second dose due to clinical deterioration, and one patient refused the second dose.

Out of 375 administered vaccine doses, 17 resulted in at least one adverse event following immunization (AEFI) being reported in the clinical diaries, yielding an overall adverse reaction reporting rate of 4.5%. Local reactions, such as injection site pain or itching, were the most common AEFIs, observed in 15 patients. Only two patients reported systemic reactions, including fever and gastrointestinal symptoms. During the observation period, no cases of HZ infection were detected among vaccinated patients.

## 4. Discussion

This ongoing initiative represents a significant step in evaluating the feasibility and effectiveness of an in-house vaccination program for oncology patients. HZ is particularly frequent among immunocompromised individuals, such as oncology patients on active treatments, due to their weakened immune systems. The RZV has been shown to be both effective and safe in reducing the incidence of HZ and its complications in this vulnerable population [14,16,17].

We prioritized HZ vaccination because of its high incidence in oncology patients and the availability of a highly effective non-live recombinant vaccine suitable for immunocompromised individuals. In contrast, pneumococcal and influenza vaccination programs are already part of standard preventive care pathways in most Italian regions.

Baseline evaluation of RZV vaccination rate in oncology patients on active treatments confirmed that coverage was much lower than recommended levels. Therefore, understanding local barriers to vaccination was important for developing and implementing a theory of change. Creating a convenient vaccination program was pivotal in reaching the project aim. Our program achieved a substantial increase in vaccination rates, rising from an initial 5.4% to 44% over the 14-month period. We believe the most important contributor to improved vaccination rates was the clear recommendation for vaccines by healthcare providers. Our staff education campaign improved vaccine prescribing patterns by disseminating guideline recommendations and encouraging staff compliance.

Additionally, engaging in vaccination efforts to limit preventable infections places some control over the cancer journey into the hands of patients and reinforces autonomy. The initiative has strengthened patient engagement and loyalty, as patients appreciate the convenience and comprehensive care offered by receiving vaccinations directly at the oncology day hospital. This study is pioneering in its attempt to integrate the vaccination process within the oncology department, allowing patients to receive their vaccines in the same setting where they receive their cancer treatments.

The scalability of this model to other cancer institutes depends on institutional resources and healthcare organization. Key facilitators include strong institutional commitment, integration of vaccination records into electronic health systems, and close collaboration between oncologists, pharmacists, and local health authorities. Common barriers involve limited staff time, lack of vaccine supply within hospital pharmacies, and reimbursement differences across regions. Educational initiatives targeting both patients and healthcare providers, along with clear communication pathways with territorial health services, can help overcome these challenges.

While the study has demonstrated a notable increase in the vaccination rate for HZ among cancer patients, it is not without limitations. Although our primary objective was to improve vaccine uptake and feasibility within the oncology workflow, early observations from routine clinical follow-up suggest a possible reduction in HZ episodes among vaccinated patients compared with unvaccinated individuals. However, formal evaluation of clinical outcomes—including HZ incidence, hospitalization rates, and treatment interruptions due to infection—is ongoing. This follow-up analysis will provide more robust evidence on the direct clinical benefits of RZV administration in cancer patients receiving systemic therapy. After 14 months, 35.7% of patients still need to be evaluated for vaccination, highlighting the difficulty of systematically identifying and assessing patients. Prioritizing patients for vaccination amid busy clinic schedules proved to be a logistical challenge.

## 5. Conclusions

In conclusion, we provide a model that can enhance overall vaccination rates and potentially improve health outcomes for cancer patients. Integrating vaccination services into the oncology day hospital, along with community-based vaccination infrastructure, represents a strategic approach for immunocompromised patients with frequent hospital visits. Lessons learned about promoting vaccination uptake may be generalizable to different populations and vaccine types.

Additionally, this patient population will be monitored in the future for any episodes of HZ despite the vaccination. This ongoing follow-up will provide further insights into the long-term effectiveness and benefits of the in-house vaccination program.

## Figures and Tables

**Figure 1 cancers-17-03502-f001:**
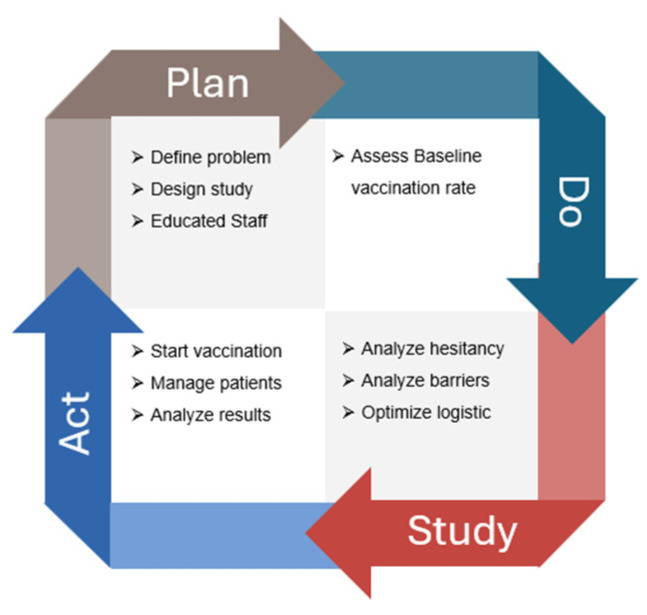
Flowchart of the quality improvement (QI) initiative aimed at increasing recombinant zoster vaccine (RZV) uptake among oncology outpatients. The figure illustrates the stepwise Plan–Do–Study–Act (PDSA) cycle applied during the project: initial assessment of baseline vaccination coverage, implementation of the in-house vaccination campaign, bimonthly monitoring of adherence, and final evaluation after 14 months. The flow highlights patient screening, eligibility assessment, vaccination delivery within the oncology unit, and outcome measurement in terms of vaccination rate improvement.

**Figure 2 cancers-17-03502-f002:**
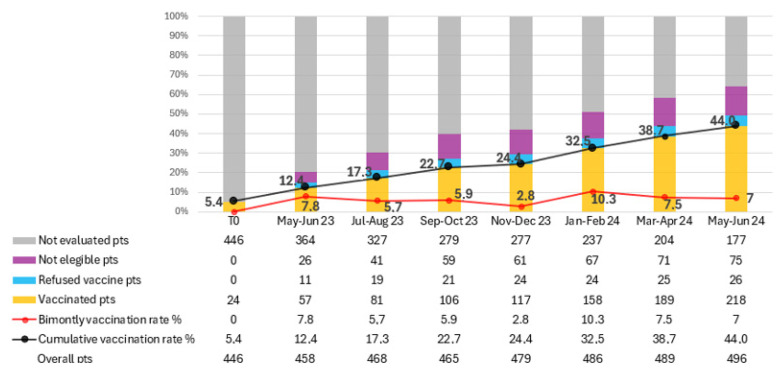
Trend in recombinant zoster vaccine (RZV) coverage among oncology outpatients over a 14-month period. The figure shows the progressive increase in vaccination adherence following the implementation of the in-house vaccination program. Data points represent cumulative vaccination rates (%) recorded every two months. The continuous rise in coverage—from 5.4% at baseline to 44% at the end of the study—demonstrates the effectiveness of the quality improvement (QI) initiative in integrating preventive vaccination within oncology care.

**Table 1 cancers-17-03502-t001:** Baseline vaccination status and previous infection in the overall cohort of oncology patients.

Questions	Respondents, *n* (%)
Previous HZ vaccination	24 (5.4)
Previous HZ infection	29 (6.5)
COVID-19 vaccination	435 (97.5)
2 doses	17 (3.9)
3–4 doses	312 (71.7)
5–6 doses	105 (24.4)
Previous COVID-19 infection	156 (35)
Influenza vaccination in the last 5 years	337 (75.6)

**Table 2 cancers-17-03502-t002:** Patients’ characteristics of new vaccinated patients.

Characteristics	Vaccinated Patients (N = 200)
**Age (years), median (range)**	71, (42–93)
**Sex, *n* (%)**	
Female	117 (58.5)
Male	83 (41.5)
**ECOG PS * at vaccine administration, *n* (%)**	
0–1	161 (80.5)
2	39 (19.5)
**Previous HZ infection, *n* (%)**	16 (8)
**COVID-19 vaccination**	193 (96.5)
**Influenza vaccination in the last 5 years**	151 (75.5)
**Cancer subtype, *n* (%)**	
Breast	55 (27.5)
Lung	44 (22)
Colorectal	27 (13.5)
Prostate	18 (9)
Bladder	11 (5.5)
Pancreas	10 (5)
Stomach	10 (5)
Gynecologic	9 (4.5)
Head and neck	4 (2)
Other	12 ^1^ (6)
**Staging TNM, *n* (%)**	
I–II	56 (28)
III	16 (8)
IV	128 (64)
**Line of therapy, *n* (%)**	
Neoadjuvant	5 (2.5)
Adjuvant	59 (29.5)
First line	110 (55)
Second line or further	26 (13)
**Treatment type, *n* (%)**	
Chemotherapy	59 (29.5)
Immunotherapy	44 (22)
Targeted therapy	40 (20)
Endocrine therapy	30 (15)
Chemotherapy + Targeted therapy	9 (4.5)
Chemotherapy + Immunotherapy	18 (9)
**Radiotherapy in the last 2 years, *n* (%)**	39 (19.5)
**Cardiovascular disease, *n* (%)**	29 (14.5)
**Diabetes, *n* (%)**	36 (18)
**Chronic steroids, *n* (%)**	25 (12.5)
**Polypharmacy (5 or more medications), *n* (%)**	70 (35)

* Eastern Cooperative Oncology Group Perfomance Status. ^1^ GIST = 3; NET = 3; Kidney = 2; Melanoma = 2; Biliary tract cancer = 2; Testicular = 1; Esophagus = 1.

## Data Availability

Data supporting the findings of this study are available from the corresponding author upon reasonable request.

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
