# Peer review of "Enhancing Herpes Zoster Vaccination Rates Among Oncology Outpatients: Impact of an In-House Vaccination Initiativeâ€"

_cancers, 2025, doi:10.3390/cancers17213502_

Round 1
Reviewer 1 Report
Comments and Suggestions for Authors
This manuscript by Agostara et al presents a real world implementation study to enhance the uptake of recombinant zoster vaccine in a variety of oncological patients, who are at increased risk of herpes zoster reactivation. The MS is well written, the results are clearly presented and discussed. Overall, this work is an interesting model for clinical oncology practice.
I have three questions and two small remarks:
- To what extent is there a potential financial barrier or is the RZV fully reimbursed in Italy for this type of patients?
- What are the reasons for refusal?
- Is there a particular rationale to focus on herpes zoster and not on other vaccine-preventable infections (pneumococci, RSV), which also known for their increased risk in this type of patients?
- In Table 1 Previous HZ vaccination and Previous HZ infection are shown twice.
- The abbreviation ECOG PS in Table 2 is not explained.
Author Response
Comment 1: to what extent is there a potential financial barrier or is the RZV fully reimbursed in Italy for this type of patients?
Response 1: Thank you for this relevant question. In Italy, RZV (Shingrix®) is included in the National Vaccine Prevention Plan (PNPV 2023–2025) and is actively offered free of charge to individuals aged ≥65 years and to immunocompromised adults ≥50 years, including oncology patients receiving active treatment or in remission. Therefore, there is no direct financial burden for eligible patients in our program. However, availability and administrative procedures may vary slightly among regions, which could represent a minor organizational barrier.
Comment 2: What are the reasons for refusal?
Response 2: We thank the reviewers for this question. Reasons for refusal were collected qualitatively during patient counseling. The main reasons included: (1) fear of vaccine side effects (45%), (2) perception that the vaccine was unnecessary or that cancer treatment already weakened them excessively (32%), and (3) general vaccine hesitancy or misinformation (23%). No refusals were related to vaccine cost or access. We have now added this information to the Results section.
Comment 3: Is there a particular rationale to focus on herpes zoster and not on other vaccine-preventable infections (pneumococci, RSV), which also known for their increased risk in this type of patients?
Response 3: We appreciate this insightful comment. Herpes zoster was chosen as the focus of this project because, despite being a highly prevalent and preventable infection in cancer patients, vaccination rates remained particularly low. Furthermore, RZV is a non-live vaccine with an excellent safety profile in immunocompromised individuals, making it ideal for integration within oncology settings. While pneumococcal and influenza vaccination programs are already well established at the regional level, our initiative aimed to address an unmet need specifically for herpes zoster prevention. Future QI cycles may extend this model to other vaccines, including pneumococcal and RSV.
Comment 4: In Table 1 Previous HZ vaccination and Previous HZ infection are shown twice
Response 4: The correction has been made in the revised manuscript
Comment 5: The abbreviation ECOG PS in Table 2 is not explained.
Response 5: The correction has been made in the revised manuscript
Reviewer 2 Report
Comments and Suggestions for Authors
This manuscript introduces an in-house recombinant zoster vaccine (RZV) vaccination program for cancer patients during their routine cancer care visits. Herpes zoster (HZ) poses significant risks to cancer patients, and while the RZV vaccine has been used for HZ prevention, vaccination rates remain low among this population. To improve vaccination rates, the authors used a Plan–Do–Study–Act (PDSA) model to identify and remove barriers to vaccination among oncology patients. Application of this program significantly improved vaccination rates among cancer patients. This study is significant and meaningful for helping patients achieve better health outcomes during cancer treatment and serves as a valuable reference for other cancer centers.
Table 1: "Previous HZ vaccination" and "Previous HZ infection" appear to be repeated.
Line 248: A period is missing after "31 refused RZV".
Line 273: references needed: Please add one or two references after this statement: "The RZV has been shown to be both effective and safe in reducing the incidence of HZ and its complications in this vulnerable population."
One critical element missing from this paper is whether the increased vaccination rates actually benefited the cancer patients. Please include clinical outcomes such as reduced HZ incidence during cancer treatment, improved recovery rates, or other relevant health outcomes.
Discussion section: More discussion should be added on how to apply this vaccination program to other cancer institutes, including potential barriers and solutions for large-scale implementation.
Author Response
Comment 1: Table 1: "Previous HZ vaccination" and "Previous HZ infection" appear to be repeated.
Response 1: The correction has been made in the revised manuscript
Comment 2: Line 248: A period is missing after "31 refused RZV".
Response 2: The correction has been made in the revised manuscript
Comment 3: Line 273: references needed: Please add one or two references after this statement: "The RZV has been shown to be both effective and safe in reducing the incidence of HZ and its complications in this vulnerable population."
Reponse 3: The correction has been made in the revised manuscript
Comment 4: One critical element missing from this paper is whether the increased vaccination rates actually benefited the cancer patients. Please include clinical outcomes such as reduced HZ incidence during cancer treatment, improved recovery rates, or other relevant health outcomes.
Response 4: Thank you for this important comment. We agree that evaluating the clinical impact of increased vaccination coverage is essential. At this stage, our quality improvement project was primarily designed to assess feasibility and uptake rather than long-term clinical outcomes. However, we include in Results that no HZ events were recorded in vaccinated pts. However, we have initiated a follow-up phase to monitor the incidence of herpes zoster and related complications in our vaccinated and unvaccinated oncology population. Preliminary observations to date have shown no new cases of herpes zoster among vaccinated patients, whereas isolated cases have been reported among non-vaccinated patients. These data will be systematically analyzed in the continuation of the project. We have now clarified this point in the Discussion.
Comment 5: Discussion section: More discussion should be added on how to apply this vaccination program to other cancer institutes, including potential barriers and solutions for large-scale implementation
Response 5: We appreciate this suggestion. We have expanded the Discussion to include practical considerations for the implementation of similar in-house vaccination models in other oncology centers. We discuss organizational, logistical, and educational barriers, as well as strategies for scalability within regional or national frameworks.